# Tenderness of the Skin after Chemical Stimulation of Underlying Temporal and Thoracolumbar Fasciae Reveals Somatosensory Crosstalk between Superficial and Deep Tissues

**DOI:** 10.3390/life11050370

**Published:** 2021-04-21

**Authors:** Walter Magerl, Emanuela Thalacker, Simon Vogel, Robert Schleip, Thomas Klein, Rolf-Detlef Treede, Andreas Schilder

**Affiliations:** 1Department of Neurophysiology, Mannheim Center for Translational Neuroscience (MCTN), Medical Faculty Mannheim, Heidelberg University, 68167 Mannheim, Germany; Walter.Magerl@medma.uni-heidelberg.de (W.M.); ethalacker@outlook.com (E.T.); vogel.simon@hotmail.de (S.V.); tklein@impp.de (T.K.); Rolf-Detlef.Treede@medma.uni-heidelberg.de (R.-D.T.); 2Conservative and Rehabilitative Orthopedics, Department of Sport and Health Sciences, Health Sciences, Technical University of Munich, 80333 Munich, Germany; robert.schleip@tum.de; 3DIPLOMA Hochschule, 37242 Bad Sooden-Allendorf, Germany

**Keywords:** hyperalgesia, long-term potentiation, deep tissue, muscle, fascia, referred hyperalgesia

## Abstract

Musculoskeletal pain is often associated with pain referred to adjacent areas or skin. So far, no study has analyzed the somatosensory changes of the skin after the stimulation of different underlying fasciae. The current study aimed to investigate heterotopic somatosensory crosstalk between deep tissue (muscle or fascia) and superficial tissue (skin) using two established models of deep tissue pain (namely focal high frequency electrical stimulation (HFS) (100 pulses of constant current electrical stimulation at 10× detection threshold) or the injection of hypertonic saline in stimulus locations as verified using ultrasound). In a methodological pilot experiment in the TLF, different injection volumes of hypertonic saline (50–800 µL) revealed that small injection volumes were most suitable, as they elicited sufficient pain but avoided the complication of the numbing pinprick sensitivity encountered after the injection of a very large volume (800 µL), particularly following muscle injections. The testing of fascia at different body sites revealed that 100 µL of hypertonic saline in the temporal fascia and TLF elicited significant pinprick hyperalgesia in the overlying skin (–26.2% and –23.5% adjusted threshold reduction, *p* < 0.001 and *p* < 0.05, respectively), but not the trapezius fascia or iliotibial band. Notably, both estimates of hyperalgesia were significantly correlated (r = 0.61, *p* < 0.005). Comprehensive somatosensory testing (DFNS standard) revealed that no test parameter was changed significantly following electrical HFS. The experiments demonstrated that fascia stimulation at a sufficient stimulus intensity elicited significant across-tissue facilitation to pinprick stimulation (referred hyperalgesia), a hallmark sign of nociceptive central sensitization.

## 1. Introduction

The most prevalent musculoskeletal pain conditions transitioning to chronicity and contributing to the global burden of disease include low back pain, neck pain and temporomandibular disorder pain [1,2]. In many cases of low back pain, no skeletal cause is obvious; thus, this type of chronic pain is often considered nonspecific. Such musculoskeletal pain conditions are favorably affected by increased bodily activity [3,4,5]. Notably, recent epidemiological studies revealed that the current global confinement situation due to the COVID-19 pandemic initiated an increased frequency of moderate exercise, lowering the prevalence of musculoskeletal pain complaints [6,7]. The development of these chronic musculoskeletal pain states is potentially linked to central sensitization, (i.e., amplified transmission at central nociceptive neurons) [8,9], which is manifested by pain hypersensitivity [10] spreading heterotopically to sites beyond those directly affected by musculoskeletal input manipulation [11,12]. Such sensitization is possibly initiated by an ongoing musculoskeletal pathology, but it is also frequently augmented or evoked by the intense modulation of sensory inputs [11,13,14] Since the fascia is densely innervated by nociceptive free nerve endings [15,16,17] an increased input originating from deep tissue (i.e., the muscle or fascia) may be associated with somatosensory changes of the overlying skin and precipitate increased pain sensitivity to nociceptive (hyperalgesia) or non-nociceptive stimuli (allodynia) [18]. Quantitative somatosensory testing (QST) enables the standardized evaluation of these manifestations as psychophysical responses to defined mechanical and thermal stimuli [19]. So far, central sensitization using high-frequency stimulation (HFS) at stimulus intensities sufficient to activate the C-fiber nociceptors was extensively studied, mostly in human skin [20,21,22,23,24,25,26,27]. We have recently shown that electrical stimulation modulates pain sensitivity in the fascia and in the muscle [11], albeit with a different direction of change and distinct pain qualities [28]. Fascia HFS facilitated pain sensitivity in the fascia itself (i.e., homotopically) is likely explained by the dense innervation of the fascia, rendering fascia HFS a strong input precipitating the fascial long-term potentiation of pain [11]. In contrast, due to sparser innervation density of the muscle, muscle HFS is a much weaker input. Thus, muscular HFS precipitates a long-term depression of pain (i.e., reduced muscle pain sensitivity) [11]. Of note, muscle HFS also depressed pain sensitivity heterotopically in the overlying fascia [11]. These bidirectional changes are indeed predicted by the general theory of bidirectional plasticity in glutamatergic synapses, which are able to form a memory-like trace leading to either long-term potentiation or long-term depression, depending on the strengths of input [29,30,31,32]. Notably, muscle HFS can reduce the pain originating from overlying fascia, suggesting a between-tissue modulation of pain sensitivity [11]. Although nociceptive muscle and skin input is differently processed at the spinal level [33,34] there is also a convergence between skin and deep tissue. Thus, like visceral input, painful muscle stimulation elicits cutaneous referred pain [35,36,37] and tenderness [36,38]. However, comprehensive QST data of the skin after the painful stimulation of deep tissue have not been provided yet. We have now approached this gap via stimulation of the multifidus muscle (MM) or the thoracolumbar fascia (TLF) with HFS of sufficient intensity to modify pain perception in these tissues [11,25], as well as another validated model, hypertonic saline injection [35,36,37]. Thus, the current study focuses on somatosensory changes of the skin after evoked central and peripheral sensitization of deep soft tissue. 

We hypothesized that painful chemical stimulation (but not electrical stimulation) of the muscle and fascia of the lower back will result in a stimulation intensity-dependent heterotopic sensitization of the overlying skin to punctate mechanical stimuli, a hallmark sign of nociceptive central sensitization [39]. Furthermore, we expected somatosensory changes of a different extent after the chemical stimulation of various fasciae. 

## 2. Material and Methods

Somatosensory evocations and changes of the skin were attempted via the electrical and chemical stimulation of the underlying thoracolumbar fascia (TLF) and multifidus muscle (MM) of the lower back. Herein, the chemical stimulation was applied using several stimulation intensities. To investigate the differences in the central nociceptive processing input of fascia tissue, the chemical stimulation protocol was further applied at other body sites, namely the temporal fascia, trapezius fascia, and iliotibial band (IT band). 

### 2.1. Participants

Forty-eight young healthy volunteers (27 female, 21 male; 24.9 ± 3.6, mean ± SD) with no history of back pain participated in this study, sixteen each for the three different experiments. All volunteers signed a written consent form and had sufficient command of the German language. The criteria for exclusion were the use of any medication or having undergone recent surgeries. None of the participants withdrew from the study prematurely. Local ethics committee approval was obtained according to the current version of the Declaration of Helsinki (Medical Faculty Mannheim, Heidelberg University ethics committee II, 2010.274N-MA, 2020-533N). 

### 2.2. Experimental Protocols

#### 2.2.1. Electrical Stimulation

Subjects were advised to lie on a bench facedown, thus minimizing back muscle contraction. Concentric bipolar needle electrodes (diameter 0.46 mm, stimulation area 0.07 mm^2^, CareFusion, USA) were positioned under ultrasound guidance (M-Turbo ultrasound system; Sonosite, Munich, Germany) bilaterally into the thoracolumbar fascia (deep fascia [40]), and into the multifidus muscle, 10 mm below the surface of the thoracolumbar fascia and approximately 2 cm lateral to the midline at lumbar level L3/L4. Electrical stimuli (pulse width: 2 ms) were applied via a constant current stimulator (DS7A; Digitimer, Welwyn Garden City, UK). Individual detection thresholds (simply noticeable sensation) were determined. Trains of high-frequency stimuli (100 Hz for 1 s) were applied at 10 times the individual electrical detection threshold and were used to elicit pain or induce long-lasting somatosensory changes of the overlying skin [10,11,20]. High-frequency stimulation (HFS) was applied through one electrode, while the other site remained unconditioned and served as a control. Before and 30 min after HFS, the quantitative sensory testing protocol (QST) was applied at the site of the overlying skin to determine somatosensory changes. The same procedure was repeated in a second session, but the HFS was applied at the side contralateral to that of the first session, stimulating the other tissue type. Electrical stimulations were each separated by at least 5 days. For more details see [11]. The experimental design of the study was a fully balanced right–left crossover design and balancing also comprised the different tissue type conditioned by HFS. Thus, all 16 participants (8 female, 8 male; 24.2 ± 2.0, mean ± SD) were blinded with regard to the stimulated tissues.

#### 2.2.2. Chemical Stimulation

In an exploratory dose-finding pilot experiment, three different amounts (50 µL, 200 µL and 800 µL) of hypertonic saline (5.8%) were injected into the multifidus muscle and the overlying thoracolumbar fascia (deep fascia [40]) at spinal levels L3/L4 in 16 healthy volunteers (11 female, 5 male; 23.3 ± 3.1, mean ± SD) to delineate adequate conditions for the stimulation intensity in order to be able to compare different tissues. Therefore, after injection, the volunteers were asked to rate the magnitude of perceived pain on a numerical rating scale (NRS) with the endpoints 0 (= no pain) and 100 (= most intense pain imaginable).

In 16 healthy volunteers (8 female, 8 male; 27.3 ± 4.1, mean ± SD), we injected 100 µL of hypertonic saline at different fascial sites to compare the impact of chemical stimulation in different fasciae. These injection sites were the iliotibial band (IT band) at a mid-distance between the greater trochanter and the fibular head at the lateral midline, the TLF approximately 2 cm lateral to the midline at the L3/L4 level overlying the multifidus muscle, and the fascia nuchae approximately 3–4 cm lateral to the midline at the C7/Th1 level overlying the trapezius muscle (all deep fascia [40]). In addition, we made an injection subcutaneously onto the temporal fascia (i.e., into the superficial fascia/superficial adipose tissue [41]). Figure 1 depicts the locations of the injection sites. The chemical stimulation of the temporal fascia/IT-band and the TLF/trapezius muscle fascia were performed during 2 different experimental days separated by at least 5 days of rest. The mechanical pain threshold (MPT; see the “mechanical testing” section) was determined before saline injection and after the pain sensation from the hypertonic saline had subsided to determine changes in pain sensitivity, which defines the magnitude of potential nociceptive facilitation related to central sensitization [39]. The position of the injection needle for each bolus injection of hypertonic saline was guided by ultrasound (M-Turbo ultrasound system; Sonosite, Munich, Germany). The experimental design was a fully balanced right–left crossover design comprised of the tissue type stimulated by hypertonic saline and the amount of saline injected. The contralateral control site remained unconditioned. 

### 2.3. Quantitative Sensory Testing (QST)

The comprehensive QST battery of the German Research Network on Neuropathic Pain (DFNS) [19,42] was performed pre- and post-HFS under identical conditions for each subject, including standardized instructions at the HFS site and the contralateral control site. One part of the QST, the mechanical pain threshold (MPT), was performed before and after chemical stimulation. The investigator was trained by a certified QST training center (certification according to the DFNS criteria, Paincert GmbH, Bochum, Germany). QST was performed in a fixed sequence as described previously [19,42]. The QST procedure started with the assessment of thermal parameters (i.e., cold detection, warmth detection, thermal sensory limen, and cold and heat pain thresholds), followed by the examination of mechanical detection and pain thresholds in the following order: mechanical detection, pricking pain threshold and suprathreshold pain rating, summation of pricking pain, vibration detection, and pressure pain threshold. The respective assessments are described in brief below. The detailed QST protocol, including reference data, has been reported elsewhere [19,43].

#### 2.3.1. Thermal Testing and Thermal Test Devices

A Peltier-based computerized thermal stimulator (TSA II, Medoc Inc., Ramat Ishai, Israel), with a 3.2 cm × 3.2 cm contact probe was used. Thermal thresholds were measured using ramped stimuli (1 °C/s) until the subject pressed a stop button signaling that they had reached the threshold for detection or pain (method of limits). All thermal ramps were started from 32 °C baseline temperatures. Cut-off temperatures for tissue safety were 0 °C and 50 °C. First, the cold and warm detection thresholds (CDT, WDT) were assessed. This was followed by a sequence of alternating warmth and cold stimuli, three of each. In this thermal sensory lime (TSL) procedure, subjects were asked to report the quality of each of the subsequent thermal stimuli. Mistaking one or more of the three cold stimuli during TSL as warm, hot, or burning pain was noted as the occurrence of paradoxical heat sensations (PHS). Afterwards, the cold and heat pain thresholds (CPT, HPT) were investigated. Here, pain indicates any “burning”, “stinging”, “drilling”, or “aching” sensation. Subjects were not permitted to look at the computer screen during thermal testing and were not given visual or auditory cues to indicate the start of the stimulus. 

#### 2.3.2. Mechanical Testing

Mechanical detection threshold (MDT) testing for A-beta fiber function (SAI slowly adapting mechanoreceptors) was assessed with a set of calibrated von Frey filaments (Optihair2, Marstock Nervtest Ltd., Marburg, Germany) that exerted forces between 0.25 and 512 mN. Using an adaptive method of limits, five ascending and descending series of stimuli applying consecutive stimulation forces were applied, and the geometric mean was calculated for the final tactile threshold.

The mechanical pain thresholds for pinprick stimuli (MPT) testing for A-delta fiber mediated hyper- or hypoalgesia to pinprick stimuli [10,44] were measured using a set of seven pinprick stimulators (with a flat contact area of 0.25 mm diameter) that exerted forces between 8 and 512 mN (The Pinprick, MRC Systems Heidelberg, Germany). Again using the adaptive method of limits, the final threshold was the geometric mean of five series of ascending and descending stimulus intensities. The subjects were instructed to discriminate whether they felt a stimulus as either “pricking” (signaling a perception above the pain threshold) or only as a tactile or blunt sensation (signaling a perception below the pain threshold). 

In a separate test, a stimulus–response function for the mechanical pain sensitivity (MPS) was determined using the same pinpricks as described above. Following each stimulus, the subject was asked to rate the magnitude of pain on a verbal numerical rating scale (NRS; 0 = not painful, 100 = maximal pain imaginable). Pain to light touch (dynamic mechanical allodynia; DMA) (which tests A-beta fiber mediated pain sensitivity to light touch) was tested via light stroking with a cotton wisp (3 mN), a cotton wool tip fixed to an elastic strip (100 mN), and a soft make-up brush (400 mN). Volunteers were asked to use the same NRS (see above). At every test site, each of the seven intensities of pinpricks and three intensities of light stroking were applied five times in a balanced order. The mechanical pain sensitivity was calculated as the geometric mean of all pain ratings for pinprick stimuli. The dynamic mechanical allodynia was quantified as the geometric mean of all numerical pain ratings after light touch stimuli. To avoid a loss of zero-values, a constant of 0.1 was added to each rating for both analyses, for mechanical pain sensitivity and for dynamic mechanical allodynia. For more details on the evaluation of MPS (see [45]).

Assessment of the stimulus response function for pricking pain was followed by an assessment of pain summation tested by the wind-up ratio (WUR). WUR was examined using a train of 10 pinprick stimuli of 256 mN (repeated at 1 Hz) and comparing the painfulness of this train of stimuli to the pain elicited by a single pinprick stimulus of 256 mN. The wind-up ratio was calculated as the mean pain rating of five series of repetitive pinprick stimuli divided by the mean pain rating of five single stimuli.

Vibration detection threshold (VDT) testing for A-beta fiber function (rapidly adapting mechanoreceptors) was examined as a flutter–vibration threshold using the semiquantitative 64 Hz Rydel–Seiffer tuning fork that has a graded readout of vibration amplitude (0–8/8 units). Vibration detection threshold was assessed with three series of descending stimulus intensities over spinous process of the lower back (L3). 

Finally, the pressure pain thresholds (PPT)—which is the only test for deep pain sensitivity in this QST battery, most probably mediated by nociceptive muscle C- and A-delta fibers—were measured using a pressure algometer (FDK20, Wagner instruments, Greenwich, CT, USA; operating range between 2 and 20 kg). The algometer had a rubber tip, with a contact area of 1 cm^2^. The algometer was pressed on the skin with an increasing ramp of 0.5 kg/s and the subject was asked to respond verbally as soon as the pressure became painful. As in all mechanical pain testing procedures, pain was indicated by any “sharp”, “pricking”, or “stinging” sensation. 

### 2.4. Statistics

QST values were analyzed as described previously [19,42,43]. To be able to compare the changes in different QST parameters beyond their different physical dimensions, data were normalized to standard normal distributions using the following transformation: individual standard z-value = (individual value − mean_baseline_)/SD_baseline_. This standardization returns dimensionless standard values of 0 ± 1 SD for the baseline data. Deviation from the baseline is thus shown in units of SD of the baseline. The sign of this transformation was chosen to give negative values for reduced sensitivity (loss) and positive values for increased sensitivity (gain). The same transformation was also used for MPT in different tissues. Statistical analysis was performed using SigmaPlot software; version 12.5 (Systat Software, Inc.) or Statistica, release 4.5 (Statsoft, Inc.). Significant differences (at *p*-values < 0.05) were determined by repeated measure analysis of variance (RM-ANOVA), followed by a Fisher–LSD post hoc test. All values given in this study are depicted as mean ± SEM. 

## 3. Results

### 3.1. Somatosensory Profile of the Skin after High Frequency Electrical Stimulation (HFS) of the Thoracolumbar Fascia (TLF) or Underlying Multifidus Muscle (MM)

Unexpectedly, the mean pain intensity elicited by focal electrical HFS stimulation in muscle or fascia was only 10.6/100 NRS and 18.0/100 NRS, respectively (data from [11]). This was about two- to fourfold lower than that of the pain induced by HFS in a common human model of the long-term potentiation of pain in the skin using an epicutaneous array of punctate electrodes, which induced a modality-specific mechanical hyperalgesia to punctate stimuli or pain to light touch [10,20,21,22,23,24,25,26]. Unlike cutaneous HFS, HFS of the TLF or the underlying MM elicited only a marginal reduction of thermal or mechanical detection never reaching statistical significance. Moreover, the pain parameters of the QST remained almost completely unchanged. The respective somatosensory profiles are shown for the fascia stimulation (Figure 2; HFS of the TLF) and for the muscle stimulation (Figure 3; HFS of the MM underlying the TLF). The profiles depict normalized changes (z-values) relative to the baseline assessed prior to HFS. Notably, the QST profiles of the HFS site never differed from the QST profiles of the unstimulated contralateral control site. None of the somatosensory tests exceeded the conventional five percent level of statistical significance. 

Moreover, subjects never reported the occurrence of thermal dysesthesia (i.e., perceiving gentle cold stimuli as “warm”, “hot”, or “burning” pain (paradoxical heat sensation (PHS))) or mechanical dysesthesia (i.e., perceiving gentle stroking tactile stimuli as painful (dynamic mechanical allodynia (DMA); data not shown)). 

CDT, cold detection threshold; CPT, cold pain threshold; HPT, heat pain threshold; MDT, mechanical detection threshold; MPS, mechanical pain sensitivity; MPT, mechanical pain threshold; PPT, deep pain sensitivity to blunt pressure; TSL, thermal sensory limen; VDT, vibration detection threshold; WDT, warm detection threshold; WUR, wind-up ratio.

### 3.2. Mechanical Pain Thresholds (MPT) after Chemical Stimulation of Three Different Volumes of Hypertonic Saline into the Thoracolumbar Fascia (TLF) or Underlying Multifidus Muscle (MM)

Assuming that the HFS of deep tissue was not sufficiently painful to induce heterotopic facilitation in the skin, we proceeded to the well-established model of chemical stimulation via hypertonic saline injection [35,36,37,38,46], which is substantially more painful than the magnitude of pain following deep tissue HFS mentioned above. In a pilot experiment, we explored multiple adjustments of stimulus intensities for the induction of central sensitization elicited by injections of different volumes of hypertonic saline (test stimulus) into the TLF and MM vs. the unconditioned contralateral control site. To this end, we assessed the pain and the ensuing changes of MPT after hypertonic saline stimulation of the TLF and MM.

Hypertonic saline injections elicited a dose-dependent pain lasting for several minutes. From the beginning, the smallest injection volume (50 µL) was about 50% more painful than HFS, namely 15.8 ± 13.9 NRS in the muscle and 25.1 ± 16.1 NRS in the fascia at peak pain (mean ± SD). The medium volume (200 µL) elicited a mean peak pain of 29.5 ± 15.7 and 45.0 ± 17.2 NRS, respectively. The largest volume (800 µL) elicited even stronger pain in both tissues, namely 35.6 ± 18.0 NRS following muscle injections and 53.9 ± 19.1 following fascia injections. In the most pain-sensitive subjects, peak pain ratings came close to or even exceeded the pain tolerance limit. Hence, pain ratings after chemical stimulations were approximately 1.5–3 times higher than pain ratings elicited by electrical HFS.

Four-way RM-ANOVA of mechanical pain thresholds (MPT) revealed that there were no robust differences between tissue types or injected volumes as a significant main effects of “tissue” (F = 2.0, *p* = 0.133) or “injected volume” (F = 0.5, *p* = 0.472), nor was there a statistically significant interaction between both (F = 2.03, *p* = 0.149).Moreover, there were no significant higher order interactions (all *p* > 0.30; Table 1). However, despite the distributed (and thus nonsignificant) main effects and interactions, post hoc analysis revealed that an injected volume of 800 µL tended to numb the cutaneous test sites significantly bilaterally (both *p* < 0.005, post hoc LSD test), while smaller injection volumes did not. Moreover, this numbness of the skin was more significantly pronounced after muscle injection compared to that following fascia injections (overall 21.7 vs. 12.7% threshold increase, *p* < 0.05, post hoc LSD), for the largest volume (with 30.2 vs. 16.6% increase, *p* < 0.01, post hoc LSD) in particular. Accordingly, we concluded that very large painful injection volumes likely obscured the potentiating effect of hypertonic saline, when injected into the muscle in particular.

From the pilot experiment, we concluded that an optimal balance of responses necessitated a sufficient level of pain to elicit the spread of hyperalgesia to adjacent tissue, (i.e., an average pain magnitude of approximately 30–40 NRS [10,20,21,22,23,24,25,26]), but it should also avoid excruciatingly high levels of pain. It should also minimize the inevitable effects of blunting the pain sensitivity via the experimental procedure. Thus, we conducted subsequent experiments comparing different test sites and aiming at potential deep tissue-to-skin facilitation using injections of a small volume (100 µL) into the fasciae rather than the muscles.

### 3.3. Mechanical Pain Thresholds (MPT) after Chemical Stimulation (Hypertonic Saline Injection) of the Temporal Fascia, Iliotibial Band, Trapezius Muscle Fascia, and Thoracolumbar Fascia

To elucidate whether deep tissue stimulations may be apt to elicit a change of pain sensitivity crosstalking into the overlying skin as either numbness or tenderness (heterotopic across-tissue somatosensory interaction), we stimulated the fascia in four different areas along the body axis (namely the more distal temporal area (temporal fascia) and lateral thigh (iliotibial band, IT band), and the more proximal areas of the trapezius muscle fascia or TLF), eliciting pain intensities approximately two times higher in overall mean compared to the HFS of the TLF. Data on pricking pain thresholds (MPT) were analyzed via RM-ANOVA (Table 2).

RM-ANOVA disclosed that the test areas differed significantly in their sensitivity to pinprick (RM-ANOVA: F_3,45_ = 4.48, *p* < 0.01). Three of the test areas (namely the temporal, TLF, and trapezius areas) exhibited almost the same MPT of approximately 35 mN and differed by <1 mN in their average MPT (all comparisons, *p* > 0.80). In contrast, the skin overlying the iliotibial band exhibited an MPT approximately 60% higher than that of any other test area (56 mN, *p* < 0.01 vs. any other test area; Figure 4A).

Moreover, there were significant trends for the “before vs. after” injection main effect (RM-ANOVA: F_1,15_ = 4.14, *p* = 0.06) and a complex three-way interaction of “before vs. after”, “control vs. test sites”, and “test areas” (RM-ANOVA: F_3,45_ = 2.24, *p* = 0.097). The complex interaction could be stepwise broken down into several differences. After injection, the injected sites exhibited an overall reduced sensitivity to pinprick (all test sites combined). However, with a closer inspection, this was only significant in the control sites, with an average increase of MPT by +20.2% (*p* < 0.01 post hoc LSD test), but it was much weaker in the test sites chemically stimulated with hypertonic saline (+7.6%, *p* = 0.22 post hoc LSD test). Comparing the test areas individually revealed that the desensitization was significant at the control site in the temporal area (+20% MPT increase, *p* < 0.05) and the TLF area (+37% MPT increase, *p* < 0.01). In contrast, there was no significant reduction of sensitivity (MPT increase) following injections into the iliotibial or trapezius areas (+14%, *p* = 0.17 and +11%, *p* = 0.46, respectively; Figure 4B).

Furthermore, in the same two areas, test sites injected with hypertonic saline were significantly more pain sensitive than their corresponding control sites after injection. In particular, the temporal test area was more pain sensitive than the control site (pain facilitation), with a pronounced relative drop of MPT (–27% compared to control, *p* < 0.001); a similar drop of MPT was also encountered in the TLF area (–18% compared to control, *p* < 0.02; Figure 4B). Baseline normalization revealed similar estimates of the threshold reduction of –26.2% (*p* < 0.001) and –23.5% (*p* < 0.05), respectively, relative to the control site. It also disclosed that local tenderness was only identified as a pain sensitivity increase when the general numbness was controlled by the assessment of MPT in unconditioned control sites (Figure 4C). Notably, the estimates of hyperalgesia to punctate stimulation (pinprick) in both hyperalgesic test areas were significantly correlated (r = +0.61, *p* < 0.005; Figure 5) and concordant in 12/16 subjects (75%). In contrast, there was no significant difference in the trapezius area (–2%, *p* = 0.66), and MPT was even increased after hypertonic saline in the iliotibial band (+14%, *p* = 0.11). Changes of MPT in these areas also did not correlate to those in the hyperalgesic areas (average r = +0.09; range: –0.08–+0.24, all n.s.). Accordingly, when successfully elicited, there was a remarkable intraindividual stability of the hyperalgesia expression and highly differentiated response patterns across the different test areas.

## 4. Discussion

The findings in the experiments reported in this paper lend further support to the hypothesis that painful stimulation in deep tissue can elicit tenderness in superficial cutaneous tissue to specific test stimuli, namely to punctate mechanical stimulation (pinprick). The elicitation of this across-tissue pain facilitation could not be easily demonstrated because it depended on the nature of the conditioning stimulus and of the stimulated deep tissue. Here, it could be elicited using a validated model of chemical stimulation (hypertonic saline) [35,36,37,38,46] of the fascia, but not of the muscle. The identification of cutaneous punctate hyperalgesia was, however, hampered by the concomitant elicitation of numbness to the same stimuli, in particular when tested following muscle injections of higher volumes. This may be due to mechanical compression evoked by the injection. However, the effect was more pronounced on the contralateral site, which can only be explained by nonlocal factors. Accordingly, it also has been reported that muscle pain induced by electrical high frequency stimulation with needle electrodes heterotopically can reduce the pain sensitivity of the fascia [11]. There is evidence that input from muscle tissue is subject to inhibitory controls. Tonic descending inhibition, activated efficiently (e.g., by ischemic muscle contractions in humans [47]), has stronger effects at the spinal dorsal horn level on neurons receiving input from deep tissues than on cells with input from the skin [48,49,50]. However, across-tissue interactions between muscle and fascia inputs have not yet been studied in detail, and thus the mechanisms of the observed inhibition remain open. Moreover, the elicitation of punctate hyperalgesia following fascia stimulation was not uniform, as it could be shown after stimulation of the temporal or thoracolumbar fascia, but not when the same stimuli were applied at the fascia of the trapezius muscle or the iliotibial band. In contrast to previous findings in the skin, focal high frequency electrical stimulation failed to elicit across-tissue sensitization, likely because it was shorter lasting, less painful, and lacking in sufficient spatial summation to constitute a strong nociceptive input necessary to cause heterosynaptic facilitation. Notably, a bias-free sorting algorithm using cluster analysis on data with experimentally induced somatosensory abnormalities (so-called human surrogate models) identified QST profiles following muscle HFS as those profiles with the lowest likelihood of somatosensory abnormality and the highest rate of matching a “healthy” (i.e., normal) profile [51]. This angle of view further supports the conclusion that muscle HFS (as well as fascia HFS), did reach the critical threshold for induction of the heterotopic modulation of cutaneous sensitivity.

Punctate hyperalgesia is the most prominent sensory sign of the central sensitization of nociceptive projection pathways [10,20,21,22,23,24,25,52], which is the nociceptive variety of synaptic long-term potentiation (LTP), the ubiquitous mechanism of the enhanced efficacy of glutamatergic synaptic transmission [9,29,30,31,32]. This facilitation comprises amplified transmission not only in the conditioned pathway (homotopic facilitation) but also in synaptically overlapping adjacent inputs (heterotopic facilitation) [20,23,39,53]. Both types of facilitation have been extensively studied in the skin as primary and secondary hyperalgesia [39,54]. The latter occurs adjacent (i.e., heterotopic) to a site of injury or strong noxious input (i.e., it occurs between different nociceptive inputs in the same tissue). However, heterotopic facilitation can also occur between nociceptive inputs originating from different tissues, and the best accepted example is the tenderness of cutaneous dermatomes originating from visceral affections [55], which is precipitated by the facilitation of cutaneous input by converging nociceptive visceral afferents. The interaction of nonvisceral deep inputs and skin do also exist, but they are less well studied.

Notably, very shortly after Clifford Woolf—based on electrophysiological animal experiments—had first proposed that central sensitization contributed to hyperalgesia [56], further experiments by his group suggested that nociceptive input from deep tissue rather than from cutaneous inputs may be more efficient to induce this synaptic facilitation in the spinal dorsal horn [33,57]. Human psychophysical data suggest that modulation of pain sensitivity between muscle and fascia is bidirectional and asymmetric, since strong muscle pain (delayed onset muscle soreness) facilitated pain sensitivity of the fascia [58], while painful fascia stimulation did not modulate muscle pain sensitivity [11]. The stimulation of nociceptive afferents of a muscle nerve facilitated nociceptive dorsal horn field potentials elicited by skin nerve stimulation, but not vice versa [57]. Moreover, muscle pain could blunt heterotopic deep pain sensitivity [59] and even the weak muscle pain induced by electrical high frequency stimulation heterotopically reduced the pain sensitivity of the fascia [11]. However, strong muscle pain and hyperalgesia failed to facilitate sensitivity in normal as well as sensitized skin [60,61].

The asymmetry of mutual facilitation between fascia and muscle can be explained by differences in their innervation. The fascia is much more densely innervated by peptidergic nociceptors, which are the decisive drivers of central sensitization [16,62,63,64,65]. These primary afferents are critical for the development of central sensitization, and their selective ablation prevents central sensitization in animals and humans [10,66,67]. Accordingly, the fascia may be a more relevant source than muscle to generate a central sensitization of pain [11], which may be one aspect of the myofascial pain syndrome [68]. Along the same line of arguments, fascial input may thus also be more prone to elicit cutaneous tenderness.

It has previously been shown that experimental and clinical pain associated with hyperalgesia was accompanied by reduced tactile sensitivity [69,70]. In particular, muscle pain induced by hypertonic saline injections inhibited touch sensitivity [71], and reduced tactile sensitivity may be associated with sensitive muscle trigger points [72]. In line with these previous reports, the results of one of our experiments showed that injections of hypertonic saline elicited a dose-dependent bilateral reduction of sensitivity to pinprick. Notably, this reduction was twice as strong in mirror-image contralateral unconditioned sites. According to its diffuse nonlocal pattern, this was likely related to the induction of endogenous pain control formerly termed “diffuse noxious inhibitory control (DNIC)”, a pain-inhibits-pain-type mechanism involving the brain stem and cortical mechanisms mitigating pain sensitivity by suppression of spinal transmission through a spinal-supraspinal-spinal negative feedback loop driven by nociceptive primary afferent input [73,74]. We concluded that hyperalgesia could only be demonstrated when it exceeded this significant concomitant reduction of pain sensitivity to punctate stimuli. These findings support the concept that the balance between diffuse and spatially nonselective pain inhibition and locally specific pain facilitation mechanisms operates to ease the local recognition of hyperalgesic areas by forming a center-to-inhibitory surround contrast [75]. The concomittant operation of two or more different processes is consistent with the general theory of sensory processes, integrating several modulation processes with potential opposing actions [76].

Interestingly, the pain inhibition was twice as strong when injections targeted the muscle compared to injections targeting the fascia. Together with a lesser capacity of the noxious muscle stimulation to induce hyperalgesia, this finding further supports the concept that the balance between induction of diffuse noxious pain-inhibiting controls (DNIC) suppressing pain sensitivity in a spatially nonselective fashion and pain-focal facilitating mechanisms differs between fascia and muscle, making the fascia a more likely source of hyperalgesia induction. Thus, affection of the muscle may be more prone to elicit numbness, while affection of the fascia leans towards an elicitation of tenderness.

However, the induction of hyperalgesia following fascial injections could not to be easily generalized, since hyperalgesia was only induced by hypertonic saline injections of the temporal fascia or thoracolumbar fascia, but not in the trapezius region or iliotibial band. Different fasciae have widely different forms and functions [77,78,79]. Thus, functional variation may be one aspect to understand the different responses to noxious stimulation. Substantial variation of the innervation pattern [80] and innervation density [17] between different fasciae has been reported. Possibly, the hyperalgesia response in the superficial leaflet of the temporal fascia and in the thoracolumbar fascia may relate to an increased nociceptive innervation and/or increased nociceptive sensitivity in these two tissues, and may also be associated with the frequent danger of the chronification of myofascial pain syndromes in these areas [81,82,83,84,85]. Notably, local injections of nerve growth factor sensitized nociceptors in the tibial fascia [86] and induced superficial cutaneous tenderness, while the same injections in the tibial muscle failed to induce this heterotopic facilitation [87]. Thus, incorporating neural innervation patterns and their wiring to central pain pathways as an additional source of variation beyond anatomically defined fascia differences may be an important feature. It is, however, still largely a terra incognita, deserving to be conquered by future research.

## 5. Conclusions

This study has shown that the chemical stimulation of deep tissues (muscle and fasciae) can cause highly differentiated response pattern across the different tissues and test areas. The experiments demonstrated that fascia stimulation (temporal fascia and thoracolumbar fascia), at sufficient stimulus intensities, elicited significant across-tissue tenderness to pinprick (applied to the overlying skin), which is a hallmark sign of heterotopic nociceptive central sensitization. Nonetheless, very large painful injection volumes likely activated the endogenous pain control system, which obscured the potentiating effect of hypertonic saline. This inhibition was twice as strong when injections targeted the muscle, compared to injections into the fascia. Together with a lesser capacity of noxious muscle stimulation to induce hyperalgesia, this finding further supports the concept that the balance between pain-inhibiting and pain-facilitating mechanisms differs between fascia and muscle, making the fascia a more likely source of hyperalgesia induction.

## Figures and Tables

**Figure 1 life-11-00370-f001:**
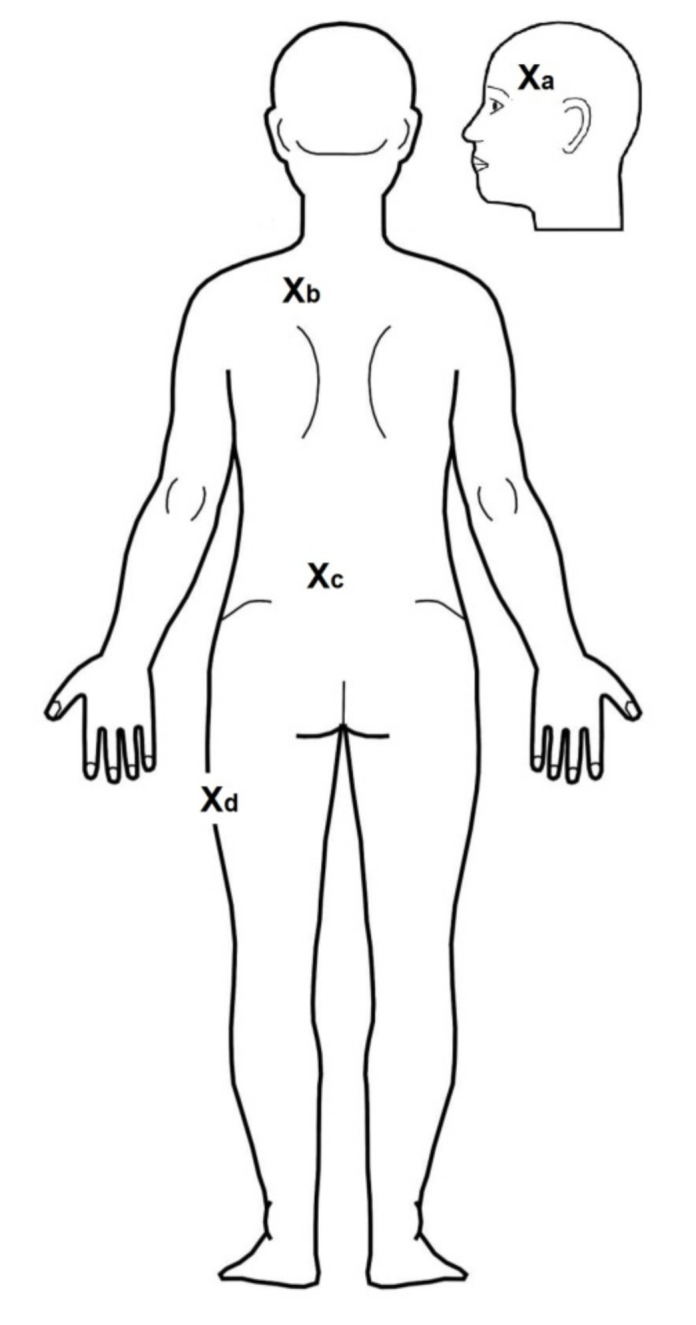
A body scheme showing the location of all stimulation sites. Xa, temporal fascia; Xb, fascia overlying the trapezius muscle; Xc, thoracolumbar fascia and the underlying multifidus muscle; Xd, iliotibial band.

**Figure 2 life-11-00370-f002:**
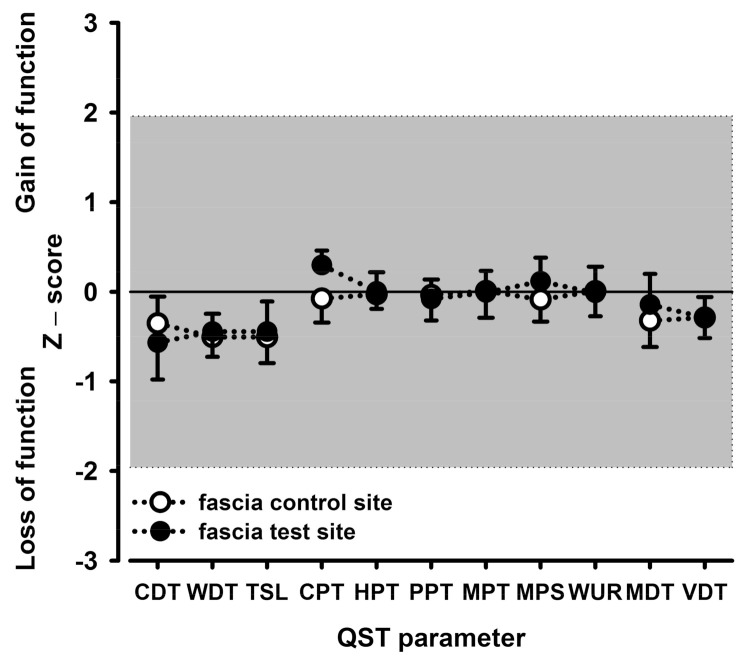
Somatosensory profiles of the skin of the lower back obtained via quantitative sensory testing (QST) following focal high frequency electrical stimulation (HFS) of the thoracolumbar fascia (TLF). The sensory profile shows thermal and mechanical detection thresholds (first and fourth block, respectively) and thermal and mechanical pain thresholds and suprathreshold pain responses (second and third block, respectively). Closed circles represent the site of the skin overlying the HFS-stimulated TLF (test site) and open circles represent the site of the skin overlying the nonstimulated contralateral TLF (control site).

**Figure 3 life-11-00370-f003:**
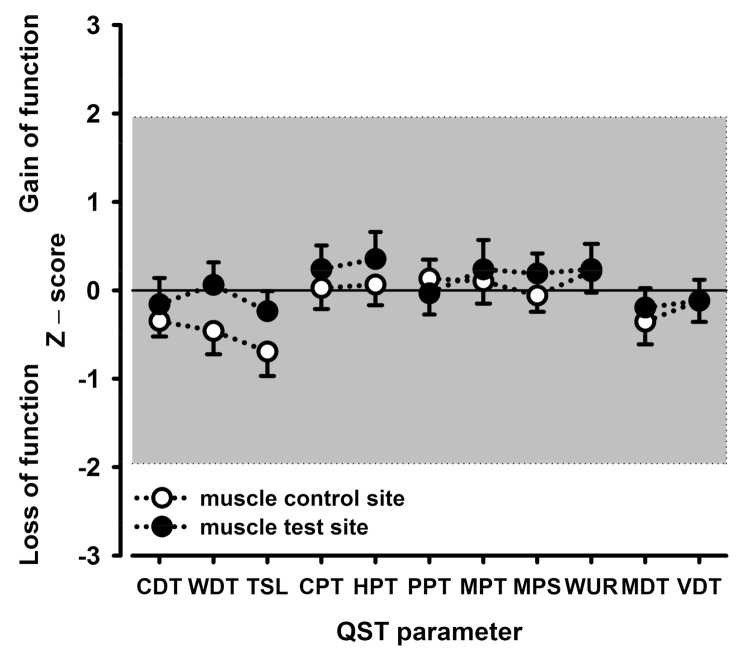
Somatosensory profiles of the skin of the lower back obtained by quantitative sensory testing (QST) following focal high frequency electrical stimulation (HFS) of the multifidus muscle. The sensory profile shows thermal and mechanical detection thresholds (first and fourth block, respectively), and thermal and mechanical pain thresholds and suprathreshold pain responses (second and third block, respectively). Closed circles represent the skin area overlying the HFS-stimulated muscle (test site); open circles represent the skin area overlying the nonstimulated contralateral muscle (control site).

**Figure 4 life-11-00370-f004:**
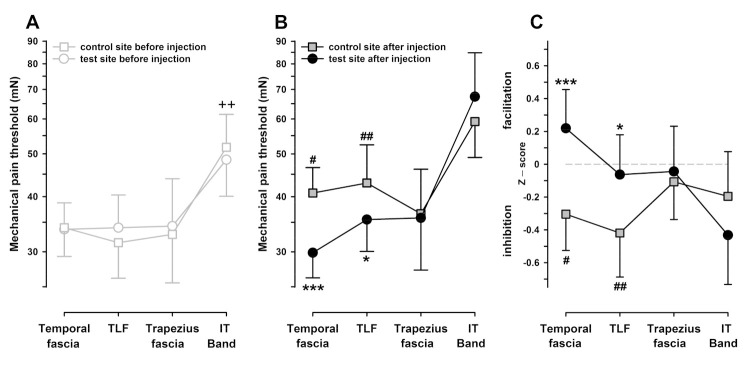
Mechanical pain threshold (MPT) in the skin overlying different fasciae injected with hypertonic saline. (**A**) Assessment of MPT before injection of hypertonic saline (5.8%) in the test sites (black circles) or unconditioned control sites (grey squares). (**B**) Assessment of MPT after injection of hypertonic saline (5.8%) in the temporal fascia, thoracolumbar fascia (TLF), trapezius muscle fascia and iliotibial (IT) band reveals that there was a generalized but variable cutaneous numbness in the unconditioned control site but not in the test site to punctate stimuli following saline injection in the test site. (**C**) Normalized changes of punctate pain sensitivity in the control and test areas (z-score values normalized to baseline MPT before injection), showing that tenderness in the test areas can be identified as relative increases of sensitivity superimposed on general numbness identified by assessment in the unconditioned control site. ^++^
*p* < 0.01 vs. all other test areas in the control and in the test sites (post hoc LSD); ^#^
*p* < 0.05, ^##^
*p* < 0.01 vs. corresponding site before injection (post hoc LSD); * *p* < 0.05, *** *p* < 0.001 vs. unconditioned control site (post hoc LSD)

**Figure 5 life-11-00370-f005:**
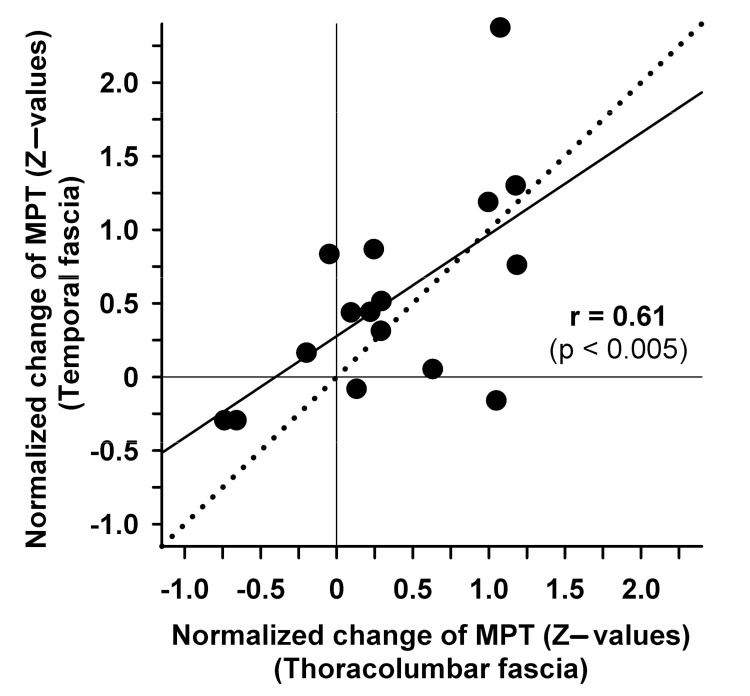
Correlation of hyperalgesic changes in mechanical pain threshold (MPT) in the temporal fascia and thoracolumbar fascia. Changes after hypertonic saline (5.8%) stimulation are shown as standard normalized data relative to the respective baselines in units of SD of their baselines (z-value). Positive values represent sensory gain (i.e., punctate hyperalgesia). Thin solid lines mark zero values (i.e., no change); the dotted line depicts the line of equality.

**Table 1 life-11-00370-t001:** Four-way repeated measures analysis of variance (RM-ANOVA) on pricking pain thresholds (MPT) before and after injection of three different volumes (50, 200, and 800 µL) of hypertonic saline (test site) injected in the thoracolumbar muscle or fascia vs. the unconditioned control site.

Effect	df_effect_, df_error_	F-Value	*p*-Value	Level of Significance
(1) Before vs. after injection	115	4.25	0.057	(*)
(2) Muscle vs. fascia	115	2,53	0.133	
(3) Test site vs. control site	115	2.01	0.176	
(4) Injected volumes	330	0.77	0.473	
1 × 2 interaction	115	2.53	0.133	
1 × 3 interaction	115	2.02	0.176	
1 × 4 interaction	115	0.47	0.631	
2 × 3 interaction	115	0.09	0.767	
2 × 4 interaction	330	2.03	0.149	
3 × 4 interaction	330	0.71	0.498	
1 × 2 × 3 interaction	115	0.60	0.810	
1 × 2 × 4 interaction	330	0.87	0.428	
1 × 3 × 4 interaction	330	0.84	0.441	
2 × 3 × 4 interaction	330	0.98	0.387	
1 × 2 × 3 × 4 interaction	330	0.03	0.968	

(*) *p* < 0.10.

**Table 2 life-11-00370-t002:** Three-way repeated measures analysis of variance (RM-ANOVA) on pricking pain thresholds (MPT) before and after hypertonic saline injection (test site) in four different fasciae test areas.

Effect	df_effect_, df_error_	F-Value	*p*-Value	Level of Significance
(1) Before vs. after injection	115	4.14	0.060	(*)
(2) Test site vs. control site	115	0.64	0.436	
(3) Test areas	345	4.48	0.008	**
1 × 2 interaction	115	1.77	0.204	
1 × 3 interaction	345	1.01	0.525	
2 × 3 interaction	345	2.24	0.397	
1 × 2 × 3 interaction	345	1.58	0.097	(*)

(*) *p* < 0.10, ** *p* < 0.01.

## Data Availability

Data can be made available by the author upon request.

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
