# Peer review of "Tenderness of the Skin after Chemical Stimulation of Underlying Temporal and Thoracolumbar Fasciae Reveals Somatosensory Crosstalk between Superficial and Deep Tissues"

_life, 2021, doi:10.3390/life11050370_

Round 1

Reviewer 1 Report

Dear Authors,

thank you for your paper, it is very interesting and covers a new topic. Before the publication, just few clarifications:

  1. can you specify the probe that you used? how Mh has it?
  2. in the temporal area, is not clear if the injection was done inside the temporal fascia or in the subcutaneous tissue (between superficial fascia and deep fascia). please clarify in line 131
  3. the use of 800 microliters of hypersaline solution can give a mechanical stimulus that can affect also the overlying tissues, please discuss this problem and how you can sure that there is no involvement of the deep fascia, when you inject such amount of liquid in the muscles

Author Response

Dear Reviewer 1,

thank you for your comments on our manuscript. We did address all your points below, followed the suggestions and dealt with them as follows:

Dear Authors,

thank you for your paper, it is very interesting and covers a new topic. Before the publication, just few clarifications:

  • can you specify the probe that you used? how Mh has it?
  • We assumed that the request aimed at the concentration of injected hypertonic saline. For clarification, we have now separated concentration and volume of hypertonic saline in the sentence of 2.2.2 paragraph 1.

  • in the temporal area, is not clear if the injection was done inside the temporal fascia or in the subcutaneous tissue (between superficial fascia and deep fascia).
  • We added some lead words to the text in section 2.2.2., paragraph 2. Please notice, that we mentioned it specifically within this sentence as follows: “In addition, we made an injection subcutaneously onto the temporal fascia (e. into the superficial fascia/superficial adipose tissue [35]).
  • please clarify in line 131 the use of 800 microliters of hypersaline solution can give a mechanical stimulus that can affect also the overlying tissues, please discuss this problem and how you can sure that there is no involvement of the deep fascia, when you inject such amount of liquid in the muscles
  • We added a new text block to the discussion in paragraph 1: The identification of cutaneous punctate hyperalgesia was, however, hampered by the concomitant elicitation of numbness to the same stimuli, in particular when tested following muscle injections of higher volumes. This may be due to mechanical compression evoked by the injection. However, the effect was more pronounced on the contralateral site, which can only be explained by other than local factors. Accordingly, it also has been reported that muscle pain induced by electrical high frequency stimulation with needle electrodes heterotopically can reduce pain sensitivity of the fascia [11]. There is evidence that input from muscle tissue is subject to inhibitory controls. Tonic descending inhibition, activated efficiently e.g. by ischemic muscle contractions in humans [47], has stronger effects at spinal dorsal horn level on neurons receiving input from deep tissues than on cells with input from e.g. the skin [48-50].

Reviewer 2 Report

Dear Authors,

Congratulations for the work done. However, I have some suggestions to improve the quality of your manuscript:

Introduction:

This section is excessively short. Due to the current context, the authors should refer to the current global confinement situation in the world with the corresponding physical health impact:

Rodríguez-Nogueira Ó, Leirós-Rodríguez R, Benítez-Andrades JA, Álvarez-Álvarez MJ, Marqués-Sánchez P, Pinto-Carral A. Musculoskeletal pain and teleworking in times of the COVID-19: Analysis of the impact on the workers at two spanish universities. Int J Environ Res Public Health. 2021;18:31-42. Doi: 10.3390/ijerph18010031.

Leirós-Rodríguez R, Rodríguez-Nogueira Ó, Pinto-Carral A, Álvarez-Álvarez MJ, Galán-Martín MÁ, Montero-Cuadrado F, Benítez-Andrades JA. Musculoskeletal pain and non-classroom teaching in times of the COVID-19 pandemic: Analysis of the impact on students from two spanish universities. J Clin Med. 2020;9:4053-4064. doi: 10.3390/jcm9124053.

Kind regards.

Author Response

Dear Reviewer 2,

thank you for your comments on our manuscript. We did address all your points below, followed the suggestions and dealt with them as follows:

Dear Authors,

Congratulations for the work done. However, I have some suggestions to improve the quality of your manuscript:

  • Introduction: This section is excessively short. Due to the current context, the authors should refer to the current global confinement situation in the world with the corresponding physical health impact:

Rodríguez-Nogueira Ó, Leirós-Rodríguez R, Benítez-Andrades JA, Álvarez-Álvarez MJ, Marqués-Sánchez P, Pinto-Carral A. Musculoskeletal pain and teleworking in times of the COVID-19: Analysis of the impact on the workers at two spanish universities. Int J Environ Res Public Health. 2021;18:31-42. Doi: 10.3390/ijerph18010031.

Leirós-Rodríguez R, Rodríguez-Nogueira Ó, Pinto-Carral A, Álvarez-Álvarez MJ, Galán-Martín MÁ, Montero-Cuadrado F, Benítez-Andrades JA. Musculoskeletal pain and non-classroom teaching in times of the COVID-19 pandemic: Analysis of the impact on students from two spanish universities. J Clin Med. 2020;9:4053-4064. doi: 10.3390/jcm9124053.

  • We added a new text block at the beginning of the introduction and included one of the suggested publications as well as another recent publication regarding the impact of the COVID pandemic to musculoskeletal pain: The most prevalent musculoskeletal pain conditions transitioning to chronicity and contributing to the Global Burden of Disease include low back pain, neck pain and temporomandibular disorder pain [1,2]. In many cases of e.g. low back pain, no skeletal cause is obvious and this type of chronic pain is often called nonspecific. Such musculoskeletal pain conditions are favorably affected by increased bodily activity [3-5]. Notably, recent epidemiological studies revealed that the current global confinement situation by the COVID-19 pandemic initiated an increased frequency of moderate exercise lowering the prevalence of musculoskeletal pain complaints [6,7].